# Color Mutations Alter the Biochemical Composition in the San Marzano Tomato Fruit

**DOI:** 10.3390/metabo10030110

**Published:** 2020-03-15

**Authors:** Gabriella Dono, Jose Luis Rambla, Sarah Frusciante, Antonio Granell, Gianfranco Diretto, Andrea Mazzucato

**Affiliations:** 1DAFNE Dept. of Agriculture and Forest Sciences, University of Tuscia, Via S.C. de Lellis snc, 01100 Viterbo, Italy; gabriella.dono91@hotmail.it; 2IBMCP Institute for Plant Molecular and Cell Biology (CSIC-UPV), Carrer de l’Enginyer Fausto Elio, s/n, 46022 Valencia, Spain; jrambla@ibmcp.upv.es (J.L.R.); agranell@ibmcp.upv.es (A.G.); 3Department of Agricultural and Environmental Sciences, Jaume I University, Av. Vicent Sos Baynat, s/n. 12071 Castellòn de la Plana, Spain; 4ENEA, Casaccia Research Center, Via Anguillarese 301, S. Maria di Galeria, 00123 Rome, Italy; sarah.frusciante@enea.it (S.F.); gianfranco.diretto@enea.it (G.D.)

**Keywords:** fruit pigmentation, introgression lines, metabolomics, mass spectrometry, San Marzano landrace, tomato

## Abstract

San Marzano (SM) is a traditional Italian landrace characterized by red elongated fruits, originating in the province of Naples (Italy) and cultivated worldwide. Three mutations, *yellow flesh* (*r*), *green flesh* (*gf*) and *colorless fruit epidermis* (*y*) were introduced into SM by backcross and the resulting introgression lines (ILs) produced the expected yellow, brown and pink fruit variants. In addition, ILs carrying double combinations of those mutations were obtained. The six ILs plus the SM reference were analyzed for volatile (VOC), non-polar (NP) and polar (P) metabolites. Sixty-eight VOCs were identified, and several differences evidenced in the ILs; overall *gf* showed epistasis over *r* and *y* and *r* over *y*. Analysis of the NP component identified 54 metabolites; variation in early carotenoids (up to lycopene) and chlorophylls characterized respectively the ILs containing *r* and *gf*. In addition, compounds belonging to the quinone and xanthophyll classes were present in genotypes carrying the *r* mutation at levels higher than SM. Finally, the analysis of 129 P metabolites evidenced different levels of vitamins, amino acids, lipids and phenylpropanoids in the ILs. A correlation network approach was used to investigate metabolite–metabolite relationships in the mutant lines. Altogether these differences potentially modified the hedonistic and nutritional value of the berry. In summary, single and combined mutations in *gf*, *r* and *y* generated interesting visual and compositional diversity in the SM landrace, while maintaining its original typology.

## 1. Introduction

In tomato (*Solanum lycopersicum* L.), fruit color is one of the most important traits affecting consumer liking, and it is the result of combined effect of carotenoids, flavonoids and eventually chlorophylls. The red color of ripe fruit comes from the accumulation of all-trans-lycopene; mutants affected in the carotenoid pathway have an altered carotenoid composition, resulting in different fruit colors [1,2]. Besides carotenoids, flavonoids play a role in determining the color of tomato fruit, particularly at the epidermal level [3]. One of the most abundant flavonoids in tomato fruit peel is the yellow-colored naringenin chalcone, often preceding and paralleling the production of lycopene in the pericarp [4]. Lastly, chlorophylls can eventually have a role in defining the color of the fruit; although they are normally degraded during ripening, “stay green” (SGR) mutants exist, that maintain important chlorophyll levels in the ripe fruit. Color variations do not only change fruit pigmentation, but they also affect the entire set of tomato fruit attributes, adversely or positively impacting the organoleptic and nutritional properties. Indeed, ripening involves a number of physiological processes ranging from the chlorophyll breakdown to the consecutive carotenoid accumulation, but also of other compounds belonging to the secondary metabolism, which play functional roles in plants [5], and have a potential in preventing diseases and promoting health in animals and humans [6,7]. Among them, acids, tocochromanols, quinones, fatty acids, sugars and polyols, as well as glycoalkaloids vary during ripening, thus modifying the nutritional value and the antioxidant activity of the fruit [8]. Three tomato mutants, *yellow flesh* (*r*), *colorless fruit epidermis* (*y*) and *green flesh* (*gf*), which are representative of the main classes of color pigments, were chosen to be studied at metabolite level, with the aim of assessing the impact of these mutations on the fruit metabolome with emphasis on compounds that co-participate in defining fruit qualities.

Yellow-fruited tomatoes have been documented since the first introduction in Europe [4,9]. The yellow color is due to the *r* mutation, represented by loss-of-function alleles of phytoene synthase 1 (*Psy1*; *Psy1* catalyzes the first rate-limiting step in the carotenoid pathway, the condensation of two molecules of geranylgeranyl diphosphate in phytoene [10]), which results in the inhibition of the whole carotenoid biosynthesis [11]. Recently, *r* tomatoes have been meeting with an increasing success, for the color novelty and the peculiar organoleptic qualities [12]. 

The *y* fruit mutant was originally described as a monogenic recessive variant leading to the formation of a colorless fruit peel [13]. The mutation, mapped on the short arm of Chr1 [14], involves the *SlMYB12* transcription factor, causing the lack of naringenin chalcone, one of the major flavonoids in tomato fruit peel, which gives the yellow color and has been proposed to influence the characteristics and function of the cuticular layer [15,16]. The pink *y*-type fruit mutation has been identified in numerous cultivated varieties that are highly consumed in Asian countries. 

Fruits of *gf* tomato mutant were described for their characteristic muddy brown color, resulting from the accumulation of lycopene coupled with the heterochronic presence of chlorophyll in the ripen fruit due to a lack of chlorophyll degradation [17]. In *S. lycopersicum*, the *Gf* locus maps on the long arm of Chr8. Further studies indicated that *gf* is a member of the SGR gene family, *SlSGR1*, a Mg-decheletase gene needed for chlorophyll catabolism [18,19]. Nowadays, many cultivated tomatoes, heirloom varieties but also modern hybrids, exhibiting the *gf* phenotype are commercially available; indeed, these varieties are appreciated and have been reported to be superior for taste-related compounds [20,21].

San Marzano (from now on referred to as SM) is a traditional variety grown in the region of Naples, Italy; it is considered an important model for fruit quality parameters, because of its intense and uniform red color, which revealed peculiar sensory profiles. Indeed SM fruits are covered by the leaves, which promotes the lycopene accumulation, but at the same time reduces the sugar content, giving birth to its typical bittersweet flavor; this is also mainly due to the ratio between the sugar content, where fructose and galactose are the main carbohydrates, and organic acids, mostly citric acid, a major factor in determining sourness, but also flavor intensity [22,23,24]. The effect of San Marzano extracts has also been the object of studies aimed at assessing nutraceutical properties [25].

The *r*, *y* and *gf* mutations, together with other variants affecting the tomato fruit phenotype, have been introgressed by backcrossing into the common genetic background SM; the single mutant lines have also been used to express two variants in double combinations [26].

In this work, we analyzed the effect of introducing color mutations in the metabolite complement of the SM fruit. Namely we compared a comprehensive set of fruit volatile compounds (VOCs), including those involved in flavor, and both non-polar (NP) and polar (P) specialized metabolites, including those involved in health properties, in the *r*, *y* and *gf* single mutant lines and their respective double mutants, to the ripe fruit of the SM genetic background. The double mutants expanded the effect on metabolism of the introduced single mutations by revealing additional additive or epistatic effects that could be further exploited for the improvement and diversification of the SM landrace by introducing innovation, while maintaining the characteristic SM typology.

## 2. Materials and Methods

### 2.1. Plant Material and Growth Conditions 

Three single mutant lines harboring *r*, *y* and *gf* and the double mutants representing all the possible combinations in the SM background were studied in comparison with the corresponding wild-type (Table 1). Details on the backcross scheme used to obtain these introgression lines, as well as the growth conditions used were reported before [26]. Briefly, eight plants per accession were transplanted and cultivated in twin rows in an unheated tunnel following standard cultural practices for indeterminate tomatoes, using tutors and weekly removal of lateral shoots. Daily temperature was controlled by a ventilation system and plants were irrigated through a drop system. The trial was repeated with identical materials and methods for two consecutive years (2017 and 2018).

### 2.2. Fruit Sampling

Before sampling, berries were visually inspected and only intact and healthy tomatoes were collected. Two biological replicates for genotype, each represented by four fully ripe berries, were harvested over a period of three days, during the first week of August, for each year. A longitudinal pericarp wedge was excised from each of the four appropriately washed berries and cut into cubes; each replica, consisting of about 30 g of fresh material, was immediately frozen in liquid nitrogen and homogenized until a fine powder was obtained. Aliquots of about 10 g of this material were freeze-dried for the analysis of non-volatile secondary metabolites. All samples, both frozen and freeze-dried, were stored at −80 °C until analysis. 

### 2.3. Volatile Detection and Quantification

For volatile analysis, two biological replicates and two technical replicates were processed and analyzed independently for the two year experiments. Prior to the analysis, frozen fruit powder (0.5 g fresh weight) from each sample was weighed in a 15 mL vial, closed and incubated at 37 °C for 10 min. Then, 1.1 g of CaCl_2_·2H_2_O and 500 µL of EDTA 100 mM (pH 7.5) were added; samples were then gently shaken and sonicated for 5 min. Then,1 mL of the homogenized mixture was transferred into a 10 mL screw cap headspace vial, where volatiles were collected by head space solid-phase microextraction as previously described [27]. A 65 µm PDMS/DVB SPME fiber (Supelco Analytical, Bellefonte, PA, USA) was used for all the analysis. Pre-incubation and extraction were performed at 50 °C for 10 and 20 min respectively. Desorption was performed for 1 min at 250 °C in splitless mode. Volatile extraction and injection were performed by means of a CombiPAL autosampler (CTC Analytics AG, Zwingen, Switzerland)). Separation and detection were performed by a 6890N gas chromatograph coupled to a 5975B mass spectrometer (Agilent Technologies Inc., Waldbronn, Germany) with DB-5ms fused silica capillary column (60 m, 0.25 mm, 1 µm) (J&W Scientific, Agilent Technologies Inc., Santa Clara, CA, USA). Oven temperature conditions were 40 °C for 2 min, 5 °C/min ramp until 250 °C and then held isothermally at 250 °C for 5 min. Helium was used as carrier gas at 1.2 mL/min constant flow. Ionization was performed by electron impact (ionization energy, 70 eV; source temperature 230 °C). Data acquisition was performed in scan mode (mass range *m/z* 35–250; 6.2 scans per second). Chromatograms and spectra were recorded and processed using the Enhanced ChemStation software (Agilent). Untargeted analysis of all the compounds in the chromatogram was performed by means of the MetAlign 3 software [28]. Compounds were unequivocally identified by comparison of both mass spectra and retention time to those of pure standards (SIGMA-Aldrich, St. Louis, MO, USA). For quantification, peak areas of selected specific ions were integrated for each compound and normalized by comparison with the peak area of the same compound in a reference sample injected regularly, in order to correct for variations in detector sensitivity and fiber aging. This reference sample consisted of a homogeneous mixture of all the samples analyzed. Data for each sample were expressed as the relative content of each metabolite compared to those in the SM reference.

### 2.4. Non-Volatile Detection and Quantification 

For both non-volatile P and NP metabolites, two biological replicates and two technical replicates, for two years, were processed and analyzed independently. Prior to analysis, 10 mg of freeze-dried fruit powder from each sample were weighed and extracted (i) with 0.75 mL cold 75% (*v*/*v*) methanol with for 0.5 mg/L formononetin as internal standard (IS) for P metabolites, as previously described [29]; and with (ii) 0.25 mL cold 100% (*v/v*), 1 mL of chloroform spiked with 25 mg/L α-tocopherol acetate as internal standard and 0.25 mL 50 mM Tris buffer (pH 7.5, containing 1 M NaCl) as described in [30]. Liquid chromatography coupled to high-resolution mass spectrometry (LC-HRMS) conditions were as previously reported for, respectively, polar [31] and non-polar [32] metabolomes. 

Metabolite identification was performed by comparing chromatographic and spectral/MS properties with authentic standards, if available, and reference spectra, and based on the *m*/*z* accurate masses as found in the Pubchem database for monoisotopic masses, or in the Metabolomics Fiehn Lab Mass Spectrometry Adduct Calculator for adduct ions. Quantification of each metabolite was carried by calculating the relative contents to the formononetin (P) and α-tocopherol acetate (NP) IS levels.

### 2.5. Statistical and Bioinformatics Analyses

Raw data were firstly inspected and manually curated for the presence of outliers (e.g., when % st.dev./avg exceed 30%). For principal component analysis (PCA), the complete dataset after log_2_ transformation and including all replicates was considered. Untargeted analysis of VOC, and NP and P metabolomes was carried out as previously reported using, respectively, MetAlign and the SIEVE software (v2.2, ThermoFisher Scientific, Waltham, MA, USA; [33]).

As untargeted analysis revealed a consistent year effect, the “Gen*Year” interaction was investigated by two-way multivariate analysis of variance (MANOVA) on those metabolites than presented less than 30% missing values. The analysis was performed with the PROC GLM procedure and the MANOVA statement implemented in the SAS software package (v9.4M6, SAS^®^ University Edition, SAS Institute Inc., North Carolina State, USA). Since “Gen*Year” interaction was found to be the less consistent source of variation, allowance was made for the existing interaction, data were mediated over the two years and all genotypes were presented with a single mean value. PCA was performed with SIMCA-P version 11 (Umetrics, Umea, Sweden) with Unit Variance normalization. The differences between each line and the SM reference were assessed using Student’s *t*-test at the 5% significance level (*p* < 0.05). Graphs were elaborated with Excel (Microsoft Office 2013, Microsoft Corporation, Washington, DC, USA). 

Venn diagrams were generated using Venny software (https://bioinfogp.cnb.csic.es/tools/venny/index.html, v2.1). Correlation networks were generated using average values over the two years under study, as previously described [34,35]. To better evaluate most robust metabolite-metabolite associations (e.g., significant correlations), the MCODE Cytoscape plugin was used [36].

## 3. Results

### 3.1. Untargeted Analysis of Volatile, Non-Polar and Polar Metabolites

Untargeted metabolomics aims to gather information on global metabolic profiles by retrieving, in an unsupervised way, as many metabolites are detectable in a GC-MS/LC-HRMS chromatograms. The comparison of the entire VOC, NP and P metabolome between the SM control and the six mutated lines detected the total features of their metabolic profile differences, setting the stage for a more specific targeted metabolomics study later in this paper. By using this approach, 263 VOCs, 746 NP and 110 P compounds were identified in the samples, many of which were differentially accumulated in, at least, one pairwise comparison (Appendix A). For VOCs, the first two principal components explained over 51% of total variance; PC1 separated *gf* and *y* from the other lines, while PC2 kept *r* plus all the combinations harboring the *y* mutation distinct from the others (Figure 1a). In Figure 1b, PCA of the first two components for the NP compounds explained almost 53% of the total variance; SM and two *green flesh* genotypes were clearly separated from *y* mutants plus *r* by the PC1, and *r* mutants were grouped in the lower quadrants by the PC2. For P metabolites, the first two PCA components explained the over 55% of total variance (Figure 1c); PC1 kept more clearly SM and the lines carrying *r* separated from *y* mutants, and *y_r* was positioned exactly halfway between its parental lines. PC2 separated lines carrying *yellow flesh*, plus *y* from SM, which was in the upper side of the graph together with *gf* and *y_gf*. Overall, the untargeted metabolomic analysis revealed that the mutations mostly affected NP rather than VOC and P metabolome, with the *r* mutant showing the highest extent of changes, including 112 NP compounds (106 down- and 6 up-regulated; Appendix A). In addition, *r* also displayed the highest epistatic attitude towards *y* (in the VOC and NP fractions) and *gf* (P metabolome), while, notably, *y* was epistatic to *gf* in the NP untargeted metabolome.

### 3.2. Estimation of “Gen*Year” Interaction in the Quantification of Targeted Metabolites

Analysis of untargeted metabolites revealed the presence of a consistent Year effect (Appendix A). Seventy-eight VOCs, 33 NP and 69 P metabolites were independently subjected to multivariate ANOVA; the interaction was not significant for VOCs, but it was highly significant for NP and significant for P compounds (Appendix A). “Gen*Year” interaction was found to be the least consistent source of variation; therefore, data were mediated over the two years and all genotypes were presented with a single mean value in targeted analyses. 

### 3.3. Targeted Analysis of Volatile Compounds 

In order to give a more specific characterization of the flavor, volatile composition of each of the mutated lines in comparison with the wild-type SM was carried out. The selected analytical strategy allowed the relative quantification of 68 VOCs unequivocally identified by both mass spectra and retention index with those of authentic standards (Appendix A). Overall, eight compounds were related to benzenoids (B), ten to branched-chain amino acid-relatives (BCAA), nine to apocarotenoids (C), two to esters (E), twenty-four to fatty acids derivatives (L), four to phenylalanine derivatives (Phe), two to sulfur compounds (S) and six to monoterpenoids (T). 

PCA of the volatile composition revealed that the first two components explained about the 54% of the total variance; the score plot showed the position of the double mutants, related to their parental lines, and with respect to SM (Figure 2a). Indeed, *r* and *y_r* were co-located in the same dial, in agreement with the VOC untargeted metabolome plot; conversely, *r_gf* placed halfway between *r* and *gf* according to PC1 (Figure 2a). Moreover, PC1 kept the mutants *r* and *y_r* separated from the other lines. PC2 placed *y_r* and its parental lines in the upper side of the graph, together with SM. The corresponding loading plot was able to identify groups of metabolites, often belonging to the same metabolic pathways, as apocarotenoids (C, in red) and lipids (L, in light blue), terpenoids (T, in green blue) and branched-chain amino acid derivatives (BCAA, in blue; Figure 2b). A comparison between the score and the loading plots revealed the overall compositional differences between the mutated lines. 

Indeed, one of the most obvious features was that all the lines harboring the *yellow flesh* mutation (*r*, *y_r* and *r_gf*) and, in a lesser extent, also the double mutant *y_gf,* were characterized by producing lower levels of volatile apocarotenoids, which was particularly dramatic in the case of some linear apocarotenoids such as 6-methyl-5-hepten-2-one (Figure 3). In the case of the mutants for *yellow flesh* this was in accordance with the scarcity of their carotenoid precursors. These lines also showed lower levels of several phenylalanine derivatives. A characteristic feature of the lines *y*, *r* and *y_r* was the high production of fatty acid derivatives together with low levels of branched-chain amino acid-related volatiles, conversely to double mutant lines *y_gf* and *r_gf*, which showed the opposite pattern. Finally, *y* and *gf* lines were characterized by higher levels of apocarotenoids and terpenoids (Figure 2b).

To further investigate the volatile compounds content of each of the mutated line in comparison with their original parental SM, a *t*-test analysis was performed. Out of 68 VOCs identified, the line with the highest number of compounds significantly different from SM was *y*, mainly because of differences in fatty acid derivatives, among other VOCs (Table 2). The lines containing the *r* mutation strongly differed for apocarotenoid volatiles, a group of metabolites considered to be involved in tomato flavor [37], representing the metabolic pathways most notably altered in these ILs (Figure 3). Some volatiles, such as the benzenoid eugenol, had higher levels in all mutant lines (Figure 3a). Mutants carrying *yellow flesh* had lower levels of the apocarotenoid β-ionone (Figure 3b), as well as 6-methyl-5-hepten-2-one, where *y_gf* acquired lower levels too (Figure 3c). Furthermore, *r*, *y* and their double mutant had higher levels of many fatty acid derivatives, such as *(E)*-2-pentenal (Figure 3d) and 1-penten-3-one (Figure 3e). Lastly, *y* was enriched in phenylalanine derivatives, such as 2-phenylethanol (Figure 3f). 

### 3.4. Targeted Analysis of Non-Polar Metabolites 

To investigate changes at the NP specialized metabolome, LC-HRMS was used to determine the level of 54 known and previously validated compounds. They were divided in different metabolic classes, including 14 fatty acids (FA), one phospholipid (PHO), two sterols (STE), 15 carotenoids (CAR), eight chlorophylls (CHL), six quinones (QUI), and five tocochromanols (TOC; Appendix A). The score plot of the first two PCA components explained about the 57% of the total variance, with double mutants differently spaced from their parental lines (Figure 4a). Indeed, PC1 kept the mutants carrying *r* separated from SM and the other lines. PC2 clearly separated *y_r* and its parental lines from the three mutants carrying *gf* plus SM, partially confirming untargeted metabolomics results. The loading plot grouped metabolites belonging to the same metabolic pathways (Figure 4b). A comparison between the score and the loading plots revealed the compositional differences between the mutated lines. Indeed, the chlorophyll group in the upper side of PC2 characterized the *green flesh* genotypes, while two quinones were in the lower side of PC2, in correspondence of the *yellow flesh* genotypes. The carotenoid group was split into early carotenoids (up to lycopene) and xanthophylls at the opposite sides of PC1; they were respectively decreased and increased in the *yellow flesh* mutant lines (Figure 4b). 

To further investigate the metabolic changes of each mutated line in comparison with the original parental SM, a *t*-test analysis was performed (Table 2; Appendix A), and we particularly focused on metabolites with sensorial (color, taste) and health-related properties. The lines carrying *r* showed the highest number of NP compounds different from SM, mainly due, as expected, to differences in carotenoids (Table 2). The lower number of differences was shown by the single mutant *y*, indicating that this genotype is more similar to SM for NP targeted compounds. 

*r*, *r_gf* and *y_r* were characterized by levels of phytoene (Figure 5a), β-carotene (Figure 5b) and lycopene (Figure 5c) lower than SM and the other lines, in agreement with previous reports; *r*, *r_gf*, *y_gf*, and *y_r* reported higher levels of the xanthophylls all-trans-neoxanthin (Figure 5d) and luteoxanthin (Appendix A). At chlorophyll metabolism level, *gf* was characterized by higher contents of both chlorophyll a (Appendix A) and b, with the latter also higher in *y_gf* and *r_gf* (Figure 5e). Moreover, *r_gf* showed higher levels of pheophytin a and pheophorbide a (Appendix A). Drawing the attention on quinones, plastoquinone increased in lines carrying *r* (Figure 5f) and plastoquinol-9 in *r* and *y_r* (Appendix A). Lastly, δ-tocopherol amount was higher than SM only in *r_gf*, while γ-tocopherol and β-tocopherol enhanced in *r_gf* and *y_gf* (Figure 5g,h). 

### 3.5. Targeted Analysis of Polar Metabolites

The relative quantification of 128 polar metabolites allowed to complete the metabolomics characterization of the six mutants under study. P metabolites were divided into different metabolic classes, including 19 amino acids (AA), 17 acids (AC), four amines (AM), two lipids (LI), one nucleic acid (NU), 15 sugars and polyols (SAP), 11 alkaloids (ALK), 55 phenylpropanoids (PHE) and three vitamins (VIT; Appendix A). The score plot of the first two components explained about the 52% of the total variance, with PC1 that particularly separated *r* and *r_gf* from all the other lines, and PC2 that identified a group including *gf* and *r_gf* together with SM (Figure 4c). Interestingly, P untargeted and targeted metabolomes differently separated the mutants under study, providing clues about a large extent of distinct metabolic components contributing to their chemical profiles. The loading plot grouped metabolites belonging to the same metabolic pathway (Figure 4d). By the comparison of the score and the loading plots, the position of some metabolites in relation to the lines studied was highlighted; indeed, many kaempferols and quercetins were in the PHE group corresponding to the area of the *y* mutants. On the contrary, many naringenins grouped in the opposite side. Most AAs were grouped together, matching with *r* and in opposition to the SAP group (Figure 4d). 

To investigate the P metabolite content of each of the mutated lines in comparison with SM, a *t*-test analysis was performed (Table 2; Appendix A), giving emphasis to nutritional- and sensorial attribute-related molecules. The line with the highest number of differentially accumulated polar compounds was the *y* single mutant, with a preponderance of down-regulated metabolites, as expected; the PHE group mostly contributed to this diversity (Table 2). On the contrary, *gf* was the line more like SM. Notably, lines containing *r* showed a higher number of AA over SM (Table 2).

As already highlighted in the corresponding loading plot (Figure 4d), lines carrying *y* had lower levels of the PHE naringenin chalcone glucoside (Figure 6a), and conversely higher levels of kaempferol-hexose deoxyhexose-pentose compared to SM, a biochemical phenotype also observed in *gf* and *r_gf* too for the latter (Figure 6b). Regarding the ALK group, calystegine B1 resulted statistically lower in all the lines, with exception of *gf* and *y_gf*, which however displayed reduced amounts compared to SM (Figure 6c). In addition, a series of primary metabolites were characterized by higher levels in the mutants under study: for example, the AA proline in *r* and *y_r* (Figure 6d) and the VIT nicotinamide in *gf*, *r* and *y_r* (Figure 6e). Similarly, the SAP glucoheptulose was higher in *gf*, whereas all the other lines were more similar to SM (Figure 6f). The LI phosphocoline displayed lower levels in lines carrying the *y* mutation (Figure 6g), while the AC sinapinic acid was higher in all lines, with the only exception of *r* (Figure 6h).

### 3.6. Bioinformatics to Investigate Metabolite-Metabolite Relationships

Bioinformatic approaches, including Venn diagram visualization and correlation network analysis, were used in order to achieve a deeper understanding of the biochemical perturbations and relationships occurring in the SM mutants under study. Venn diagrams showed the degree of overlap for VOC, NP and P metabolites in each double mutant and its two parental single mutants. For line-specific metabolites in each group, all lines, except *y_gf* and *y_r*, showed a higher number of significantly up-regulated metabolites (Figure 7). When the overlaps between single mutants and the respective double combinations were considered, the epistasis of *r* over *gf* (29 metabolites were in common between *r* and *r_gf*) and *y* (28 metabolites in common between *r* and *y_r*) and of *y* over *gf* (21 metabolites in common between *y* and *y_gf*) were found (Figure 7).

Furthermore, we used a correlation network approach to investigate mutation-induced alteration at VOC, NP and P metabolome levels. To this purpose, three networks were built by integrating all differentially accumulated metabolites in, at least, each single mutant and the corresponding double mutants (Figure 8). Overall, the three force-directed networks allowed the achievement of specific topologies according to the distribution of the significant correlation networks existing in each metabolite-metabolite interaction; notably, a high extent of conservation was observed in each network, either in the direct (PHE in the *y*- and CAR in the *r*-yielding networks) or not direct (L volatiles in *r, y* and *r_y;* and AA in *r*, *gf* and *r_gf*) targets of the mutations. In order to evidence the more robust and strongest correlations, the MCODE Cytoscape plugin was applied to each of the three networks (Appendix A). In this way, it was possible to identify the highly interconnected regions, which resulted particularly abundant in the *r*/*gf* mutants, with a lower number of greatly dense clusters including primary (SAP, AA) and secondary (isoprenoids as CAR, CHL, QUI, T and PHE) compounds; on the contrary, the *y*/*r* mutants were characterized by a higher number of clusters with lower density, and with a high representation of amino and organic acids, besides CAR and PHE.

## 4. Discussion

This study focused on the analysis of the volatile and non-volatile compounds of six tomato lines introgressing mutations that have been selected as representative for affecting the main classes of fruit pigments. Interest towards these mutations is proved by the increasing introduction of cultivars with yellow, pink and brown fruits in the fresh tomato market and by scientific studies addressing the properties of single [38,39] and multiple [40] mutants. The characterization of the biochemical effects of these mutations was based on the comparison with the SM original parent, with the final aim of defining whether the lines showing aesthetic novelty were also diversified for organoleptic and nutritional characteristics. 

When the studied compounds were considered all together, *y* showed the highest level of variation. Venn diagrams indicated epistasis of *r* over both *gf* and *y*, as highlighted by the univariate analysis. Interestingly, when integrated and subjected to a correlation network analysis, differentially accumulated metabolites in any of the six single and double mutant ILs exhibited a strong level of coordination: indeed, irrespectively to the metabolic class object of the mutation, metabolites acting in the same pathway clustered together, thus indicating a great conservative capacity of the fruit metabolism in its mutation-derived reorganization. This finding is consistent with previous reports showing, either in tomato and grape, a general phenotypic-metabolic plasticity in response to genetic or environmental changes [24,30,41]. However, looking at the sub-clusters generated by each network, distinct and specific relationships were unraveled, with the mutants carrying the *y* and *r* mutations involving the largest number of metabolites belonging to highly diversified pathways.

### 4.1. Biochemical Changes in Fruits of Yellow Flesh and of Its Combinations with y and gf 

Mutants containing *r* were clearly separated from the other lines by NP targeted metabolites, mainly due to the compositional differences of carotenoids. The double mutant *r_gf* mapped in a distinct position compared with *r* and *y_r*, indicating partially additive and partially synergistic effects of this combination. The primary effect of the *r* mutation in the carotenoid pathway was to strongly reduce phytoene, and the colored carotenoids β-carotene and lycopene, as expected [42]. Phytofluene was also reduced in *r* genotypes, as previously described in tomato accessions [43] and in ripening fruits of *Psy1* knockout lines [44]. In the latter case, a concomitant decrease in the volatile apocarotenoids was also observed [44]. In literature, the study of *r* metabolites was often limited to the analysis of carotenoids, including the xanthophyll lutein, whose levels, in line with our results, did not vary [11]. Intriguingly, we evidenced that all-trans-neoxanthin, the last xanthophylls in the carotenoid pathway, in *r* mutants showed levels higher than SM; it is possible that this represents a cellular strategy implemented to compensate for the decrease of other carotenoids. Indeed, xanthophylls act in flowing the energy through the photosynthetic apparatus and protecting organisms against damage caused by photosynthesis itself [45]. In addition, both violaxanthin and neoxanthin are key substrates in ABA biosynthesis, and their variation can influence fruit ripening and the attitude towards abiotic stresses [46]. In this context, enhanced levels of xanthophylls could provide enough metabolic flux to guarantee an adequate ABA production. The same hypothesis can be extended to the analysis of quinones, such as plastoquinone and plastoquinol-9, whose levels increased in *yellow flesh* mutants. The importance of quinones in basic metabolic processes, such as respiration and photosynthesis, has been established [47] and an increase in their content is plausible, in a scenario where it is necessary to compensate for the lack of carotenoids. 

Important metabolic changes occurred in the *r* mutants for P compounds; several amino acids positively varied, as proline in the single mutant and in combination with *y*, and valine/norvaline in the single mutant or combined with *gf*. Proline and valine have significant functions in plant cells as plant stress sensor, ABA and polyamines interactors, and precursors of BCAA-derived volatiles [48,49]. Similarly, the vitamin nicotinamide, that plays a primary role in pyridine metabolism [50], also increased in *r*, alone or with *y*. 

Carotenoid-derived volatiles have an important role in tomato flavor, as their levels positively correlate with tomato acceptability [51]. Loss-of-function of *Psy1* in *yellow flesh* mutants led to the lack of substrates for apocarotenoid production [52], thus justifying the low amount of C VOCs such as β-ionone [53] and 6-methyl-5-hepten-2-one [54]. Conversely, the L 1-penten-3-one and *(E)*-2-pentenal increased in *r* and *r_y*, suggesting a role for the lipoxygenase–linoleate (LOX) enzyme that catalyzes the oxidation of polyunsaturated fatty acids by molecular oxygen with the formation of unstable hydroperoxides which in turn oxidize carotenoid pigments [55].

In summary, compensating the depletion of the principal carotenoids, lines carrying the *r* mutation offer peculiar nutraceutical properties, for an increased content of amino acids, vitamins, xanthophylls and quinones. Xanthophylls have been highlighted in recent studies for their positive contribution towards total dietary carotenoid intake [56], whereas quinones showed positive properties in treating cardiovascular diseases, chronic gingivitis and periodontitis and a favorable impact on cancer treatment and human reproductive health [57]. Differently, tomato *r* lines will probably show a lower score for the aroma, considering that some of the apocarotenoids providing floral or fruity notes are less represented in yellow fruits.

### 4.2. Biochemical Changes in Fruits of Colorless Fruit Epidermis and of Its Combinations with r and gf 

Multivariate analysis of targeted P compounds tightly grouped lines carrying the *y* mutation, an expected result as *y* is characterized by substantial modifications of the class of phenylpropanoids, mainly due to the lack of yellow pigment naringenin chalcone [15]. As also observed in the case of the NP fraction, untargeted and targeted metabolomes did not perfectly matched, suggesting the existence of larger, still unexplored, metabolic changes. The decrease or absence of naringenins also characterized *y* and its two double mutants, leading to the conclusion that *y* is epistatic on *gf* and *r* for this class of compounds. Compared to red SM fruits, kaempferol-hexose-deoxyhexose-pentose was higher in *y* mutants, even if many other kaempferols showed no differences. Similarly, several quercetins showed higher levels in lines carrying *y*. Variations in the PHE pathways could explain the reduced phosphocholine levels detected in the three *y* mutants, since increased activity of pathways generating cuticular lipids in tomato fruit peel precedes that of phenylpropanoid and flavonoid biosynthesis pathways [58]. 

No substantial differences in carotenoids were found between *y* and SM, except the higher all-trans-neoxanthin levels in *y_gf* and *y_r*. Conversely, tocochromanol metabolism was altered in *y* mutant berries as there was a decrease in tocopherols, mainly affecting *y* and *y_r*, confirming previous reports [15,59].

Among VOCs, differences were found in compounds derived from phenylalanine, such as eugenol and 2-phenylethanol, in line with the known alterations of *y* in the phenylpropanoid branch of phenylalanine catabolism [54]. Indeed, a significant number of VOCs derived from amino acids, are considered to have an influence on tomato flavor and liking, such as guaiacol and eugenol. In addition, 2-phenylethanol is a main contributor to tomato flavor, increasing floral aroma and the perception of sweetness [60,61], it is plausible that increase of these class of volatiles exerts a positive effect on the aroma in lines carrying the *y* mutation. In addition, *y* mutants were characterized by higher production of many fatty acid derivatives, which are the most abundant volatiles produced in the tomato fruit [62]. These VOCs include several C_5_ compounds such as 1-penten-3-one or (E)-2-pentenal, and C_6_ volatiles such as 1-hexanol, (*Z*)-3-hexenal, (*E*)-2-hexenal, or hexanal, among others, that are classified as “green leaf” volatiles due to their ‘green’ characteristic, with a fresh aroma of cut grass. Although some studies suggested a reduced impact on tomato flavor and no effect on consumer preference of these compounds [63,64], others claim their potential impact on overall flavor intensity and liking [37].

In addition to variation of the aroma, *y* mutants are endowed of higher kaempferol and quercetin levels, but also present the drawback of lower naringenin and tocopherols. Since all tocochromanols are potent lipid-soluble antioxidants and are essential dietary nutrients for mammals as vitamin E [65], this aspect represents a negative trait in *y* tomato lines. 

Therefore, the *y* mutation is to be considered as a source of more aesthetic (color) and organoleptic (aroma) than nutritional novelty. Although no clear relationship between colorless epidermis and fruit shelf-life, peculiar mechanical properties of the *y* epicarp were manifested by the fact that the peel of the mutant fruit was richer of lignin [15,66]. Intriguingly, the SM lines carrying *y*, alone or in combination, showed a higher resistance to storage, indicating that pigment variation in the peel implicates different mechanical properties and post-harvest behavior of the fruit [26]. 

### 4.3. Biochemical Changes in Fruits of Green Flesh and of Its Combinations with r and y 

The group of *green flesh* mutants showed no changes in carotenoids, but as expected, it was primarily affected in the content of chlorophylls. *gf* belongs to the so-called cosmetic-type SGR mutants, that have been described in several plant species to retain substantial amount of chlorophylls in fruits during ripening, maintaining other ripening-related metabolites, such as lycopene, unchanged [67]. Accordingly, all the SM lines containing *gf* showed fruits with higher amounts of chlorophylls. Among other NP metabolites, several tocopherols increased in *gf* single and double mutants, confirming previous data that showed higher tocopherol levels in *gf* [68,69] and *r_gf* (A. Mazzucato and G.P. Soressi, unpublished results) mutants.

For P metabolites, *gf* was similar to SM, with few notable exceptions involving flavonoids, such as kaempferols, and sugars, such as glucoheptulose. 

No substantial variations were found for *gf* mutants in VOCs except for a high eugenol content that was paralleled by the other lines. 

Because of the positive effects of chlorophylls and chlorophyll-related metabolites on cellular inflammation and as anti-mutagen and anti-carcinogen agent [70,71], and of the antioxidant function of vitamin E-producing metabolites, that reduce free-radical damage to membrane lipids by scavenging peroxyl radicals [72,73], tomato lines carrying *gf* may be considered an alternative to red tomatoes offering added nutraceutical value. In addition, *gf* may play important roles in enhancing organoleptic qualities. Although a direct, positive effect of *gf* on soluble solids content has not clearly been established, an SlGLK2-enhanced chlorophyll content in immature green fruits led to an increment in total soluble solid content in ripe fruits, possibly by the positive regulation of sugar metabolism enzyme-encoding genes [69]. This effect will be additive to that of other taste-related compounds, such as glutamate [20]. In our *gf* lines, glutamic acid was not significantly different from SM, but, considering only data in one year, it was substantially higher in all lines. All these properties justify the interest towards breeding new *gf* tomato lines and the present success in the market promises a further development in the near future.

## 5. Conclusions

This work was based on the study of the effects of mutations with an emerging commercial interest, compared within the traditional Italian tomato variety San Marzano. This biochemical and bioinformatics characterization has given further insights on the effect of each mutation on fruit aesthetic, flavor and nutritional composition. The analysis of the respective three double mutants offered an added value, making it possible to establish epistatic or synergistic effects between each pair of mutations and represented a starting point for breeding new tomato lines with different phenotypes. Although *r* and *y* cause the decrement of the most important classes of health-related pigments (carotenoids and flavonoids), the compensating increase of other metabolites with nutraceutical (xanthophylls, tocopherols, amino acids) or flavor-related (phenylalanine and fatty acid derivatives) positive properties make the studied lines worthy of attention for breeding novel and better tomatoes. Novel phenotypes could take advantage of new breeding techniques as genome editing to recapitulate the original mutations in different tomato backgrounds, or in different species, and obtain the wanted variation directly avoiding backcrossing and linkage drag effects. Ultimately, advanced breeding programs will convert the new lines into novel elite varieties of commercial interest.

## Figures and Tables

**Figure 1 metabolites-10-00110-f001:**
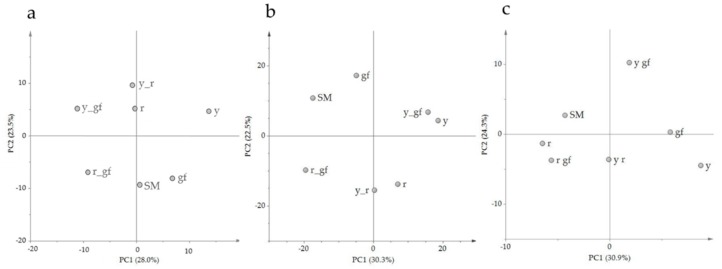
PC1 × PC2 score plots of the six mutated lines plus San Marzano (SM) according to relative values of 263 VOCs (**a**), 746 NP (**b**) and 110 P (**c**) metabolites measured by GC-LS and LC-HRMS. Line symbols are explained in Table 1.

**Figure 2 metabolites-10-00110-f002:**
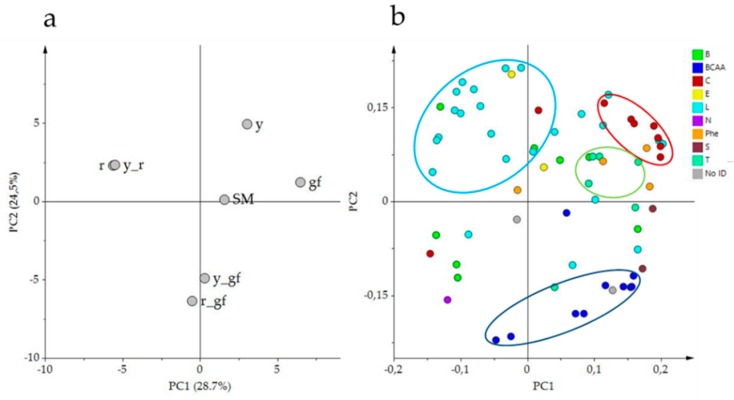
PCA of log2 values of 68 volatile compounds measured by a solid-phase micro-extraction gas-chromatography coupled to mass spectrometry (HS-SPME/GC-MS). (**a**) PC1 × PC2 score plot of the six mutated lines plus San Marzano (SM). (**b**) PC1 X PC2 loading plot. Line (a) and metabolite class (b) abbreviations are explained in Table 1 and Table 2 respectively.

**Figure 3 metabolites-10-00110-f003:**
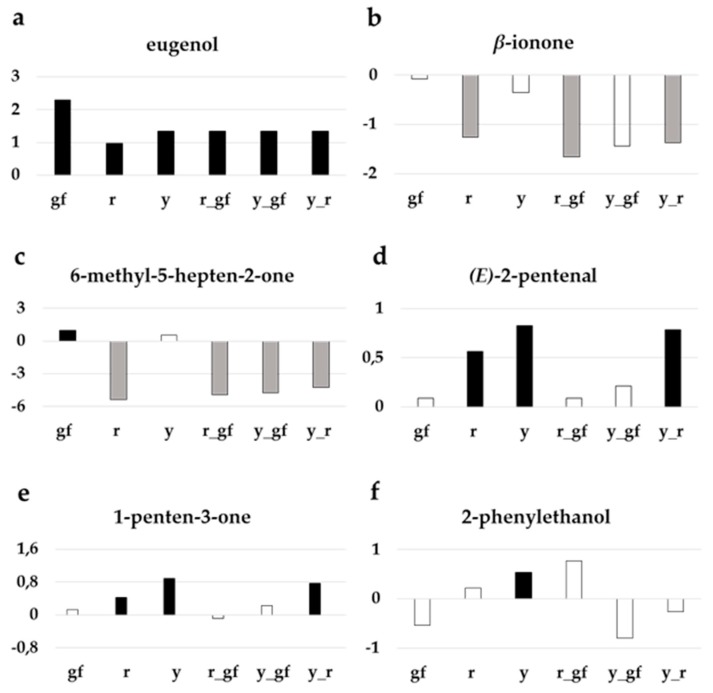
Relative log2 variation in selected volatiles involved in tomato flavor: (**a**) eugenol, (**b**) *β*-ionone, (**c**) 6-methyl-5-hepten-2-one, (**d**) *(E)*-2-pentenal, (**e**) 1-penten-3-one, (**f**) 2-phenylethanol in the six fruit mutant lines in San Marzano (SM) background compared with the recurrent parent. Line symbols are reported in Table 1. Bars colored in grey and black indicate means significantly lower and higher than SM for *p* ≤ 0.05 after Student’s *t*-test, respectively.

**Figure 4 metabolites-10-00110-f004:**
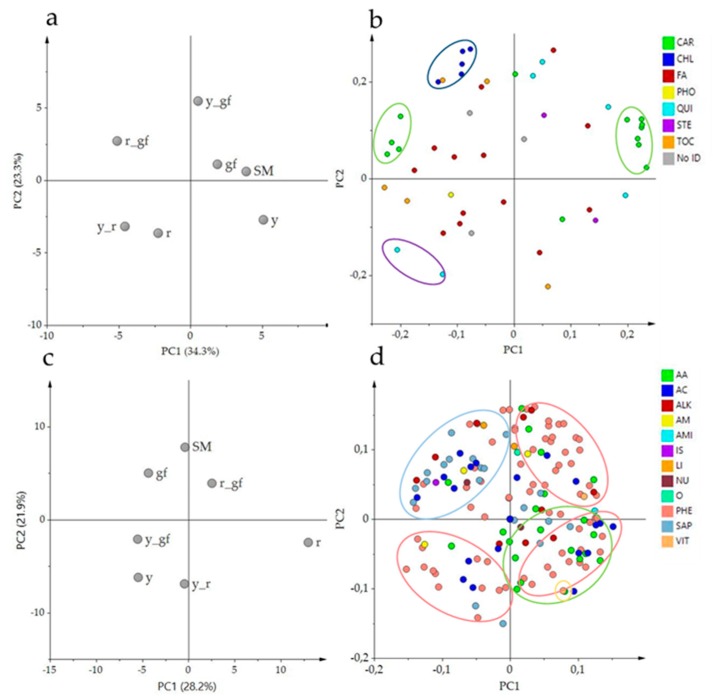
PCA of log2 values of 54 NP (**a**,**b**) and 128 P (**c**,**d**) metabolites measured by LC-HRMS. (**a**–**c**) PC1 × PC2 score plot of the six mutated lines plus San Marzano (SM). (**b**–**d**) PC1 × PC2 corresponding loading plots. Line (**a**,**c**) and metabolite class (**b**,**d**) abbreviations are explained in Table 1 and Table 2 respectively.

**Figure 5 metabolites-10-00110-f005:**
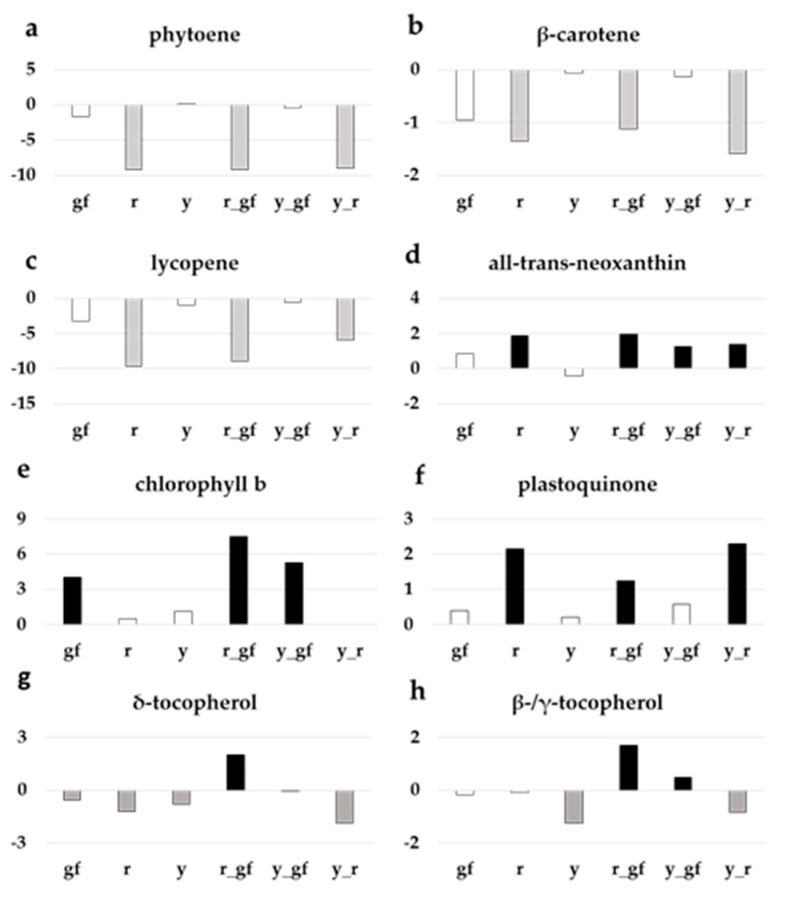
Relative log2 variation in (**a**) phytoene, (**b**) β-carotene, (**c**) lycopene, (**d**) all-trans-neoxanthin, (**e**) chlorophyll b, (**f**) plastoquinone, (**g**) δ-tocopherol, (**h**) β-/γ-tocopherol, of six fruit mutant lines in San Marzano (SM) background compared with the recurrent parent. Line symbols are reported in Table 1. Bars colored in grey and black indicate means significantly lower and higher than SM for *p* ≤ 0.05 after Student’s *t*-test, respectively.

**Figure 6 metabolites-10-00110-f006:**
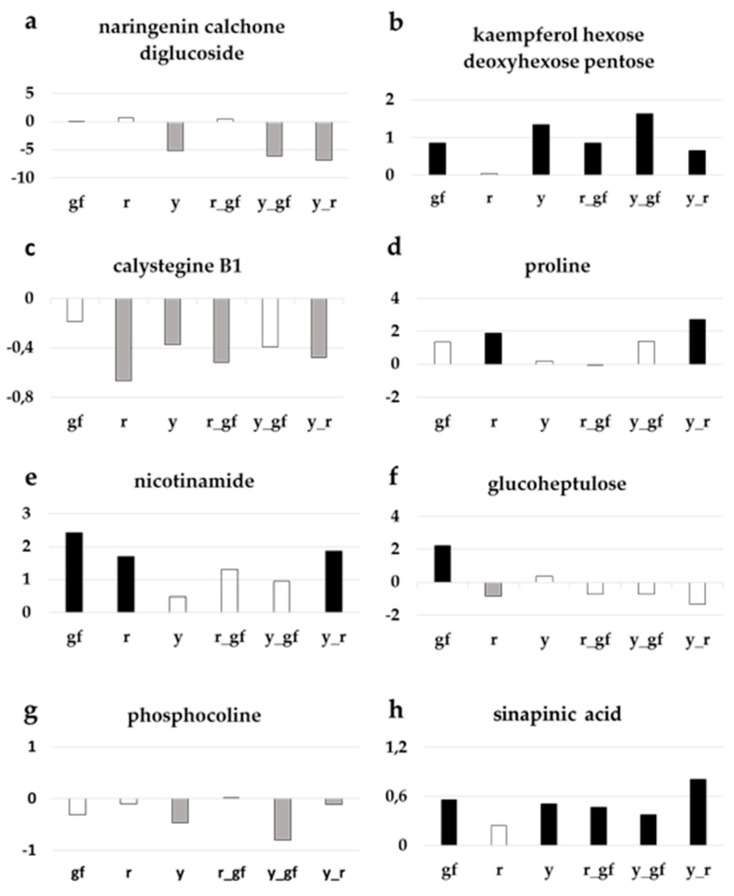
Relative log2 variation in (**a**) naringenin chalcone glucoside, (**b**) kaempferol-hexose-deoxyhexose-pentose, (**c**) calystegine B1, (**d**) proline, (**e**) nicotinamide, (**f**) glucoheptulose, (**g**) phosphocoline, (**h**) sinapinic acid, of six fruit mutant lines in San Marzano (SM) background compared with the recurrent parent. Line symbols are reported in Table 1. Bars colored in grey and black indicate means significantly lower and higher than SM for *p* ≤ 0.05 after Student’s *t*-test, respectively.

**Figure 7 metabolites-10-00110-f007:**
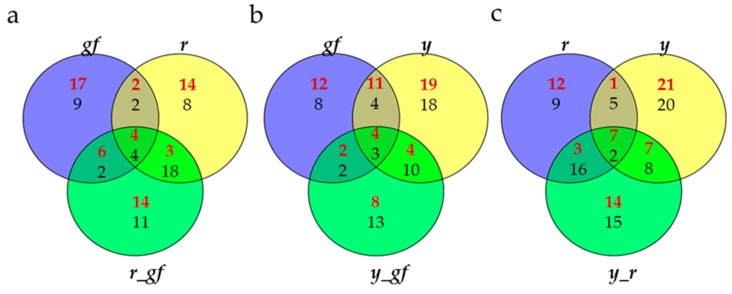
Venn diagrams for (**a**) *gf*, *r* and *r_gf*; (**b**) *gf*, *y* and *y_gf*; (**c**) *r*, *y* and *y_r*. Red and black numbers correspond to up- and down-regulated metabolites, respectively**.**

**Figure 8 metabolites-10-00110-f008:**
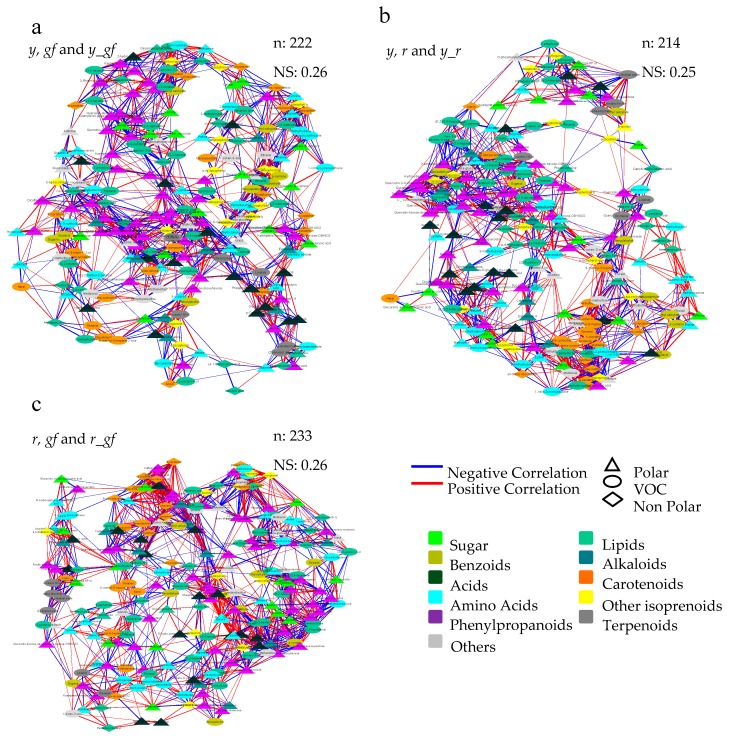
Correlation networks of metabolomics data in (**a**) *y, gf* and *y_gf*; (**b**) *y, r* and *y_r*; (**c**) *r, gf* and *r_gf* mutants. All the data from volatile (VOC), non-polar (NP) and polar (P) analyses were used as fold change to the San Marzano (SM) reference. Network topology was directed by the force of Pearson correlation coefficient index (Appendix A). Each node represents VOC (turquoise circle), a NP (diamond) or a P (triangle) metabolite. Lines joining the nodes represent positive (red) or negative (blue) correlations. Node sizes are proportional to the respective node strengths, which are shown in Appendix A. Node color is depending on the metabolic class of each compound as indicated in the figure. Number of nodes (n) and network strength (NS) are shown at the top of each network. Only correlations with |ρ| > 0.95 are shown (*p* ≤ 0.05).

**Table 1 metabolites-10-00110-t001:** Extended names, genetic symbols and description of the berry color of the seven studied genotypes, including San Marzano (SM), three single mutant and their respective double mutants.

Class of Material	Class of Variation	Name	Genetic Symbol	Fruit Color
Wild-type	-	San Marzano	SM	Red
San Marzano fruit variants	Chlorophyll	*green flesh*	*gf*	Muddy brown
	Carotenoids	*yellow flesh*	*r*	Yellow
	Flavonoids	*colorless fruit epidermis*	*y*	Pink
	Double mutants	*yellow flesh + green flesh*	*r_gf*	Light yellow
	-	*colorless fruit epidermis + green flesh*	*y_gf*	Wine-coloured
	-	*colorless fruit epidermis + yellow flesh*	*y_r*	Green

**Table 2 metabolites-10-00110-t002:** Number of compounds in the different categories and classes of metabolites that are significantly different from San Marzano (SM) in each of the six lines under study.

Metabolomics Fraction	Metabolic Class	Abbreviation	No. of Compounds	No. of Differentially Accumulated Compounds Over SM					
				*gf*	*r*	*y*	*r_gf*	*y_gf*	*y_r*
Volatile	Benzenoids	B	8	2	2	3	2		3
	Branched-chain amino acid derivatives	BCAA	10	3	3	2	4		1
	Apocarotenoids	C	9	3	5	3	6	3	5
	Esters	E	2						
	Fatty acid derivatives	L	24	3	4	6	1		4
	Others	Phe, S, T, No ID *	15	6	4	8	4		6
**Total**		**VOCs**	68	17	18	22	17	3	19
Non-polar	Carotenoids	CAR	15	2	5	3	7	5	7
	Chlorophylls	CHL	8	2	1		2	3	
	Fatty acids	FA	14	1				1	
	Phospholipids	PHO	1			1	1		
	Quinones	QUI	6	1	2		2		3
	Tocopherols	TOC	5	1	1	1		1	2
	Others	STE, No ID *	5	1	1		1		
**Total**		**NP**	54	8	10	5	13	10	12
Polar	Amino acids	AA	19	2	6	3	4	3	6
	Acids	AC	17	5	5	6	5	4	4
	Amines	AM	4			1	1		1
	Sugars and polyols	SAP	15	3	3	3	2	4	2
	Alkaloids	ALK	11	1	4	2	3	1	2
	Phenylpropanoids	PHE	55	6	13	24	18	18	16
	Vitamins	VIT	3	1	2				1
	Others	A, NU, LI, No ID *	5			1		2	1
**Total**		**P**	129	18	33	40	33	32	33
**Gran total**			251	43	61	67	63	45	64

* Undefined compounds.

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
