# Peer review of "Color Mutations Alter the Biochemical Composition in the San Marzano Tomato Fruit"

_metabolites, 2020, doi:10.3390/metabo10030110_

Round 1

Reviewer 1 Report

The manuscript is the continuation of authors recently published work on generation and characterization of genetic variants in the San Marzano background. San Marzano is a famous red elongated fruit tomato landrace which has a market value both as fresh and processing tomatoes. Previously, they have reported vegetative, reproductive and fruits traits and in this manuscript present interesting biochemical analyses of single and double mutants for fruit color (muddy brown, yellow & pink). Untargeted and targeted analyses of volatiles, non-polar and polar metabolites and correlations were performed by using latest techniques and tools.

This manuscript has many strengths:

High impact of this research on development of novel tomatoes with nutritional value.

Valuable new information.

Research design and execution is very good.

The manuscript while written as highly descriptive and informative, however, lacks the scholarship of synthesis of information obtained. The Discussion contains too much repetition of the other text and is not scholarly. It needs to be re-visited and re-written, particularly focusing on the significance of the data analyzed in Results section. It would be worthwhile to include in the Discussion section the relationship of each mutant line to the specificity of which metabolite(s) being affected and the logical conclusions.

A lot of descriptive parts could be minimized, and the focus be given to the functional possibilities/relationships of the specific mutant lines to the metabolic pathway(s) impacted. This can then be translated into the significance of these studies for nutrition and other aspects of ripening processes. For instance, what evidence is there for the authors to suggest that the enhanced ripening causes membrane breakdown and thereby release of some of the molecules the authors quantified, A deeper insight of the data presented is missing. 

In its present narrative, this manuscript is purely descriptive and lacks scholarship in terms of condensed thinking about metabolic regulation vis a vis each mutation.

Author Response

Reviewer n. 1

“The manuscript while written as highly descriptive and informative, however, lacks the scholarship of synthesis of information obtained. The Discussion contains too much repetition of the other text and is not scholarly. It needs to be re-visited and re-written, particularly focusing on the significance of the data analyzed in Results section. It would be worthwhile to include in the Discussion section the relationship of each mutant line to the specificity of which metabolite(s) being affected and the logical conclusions. A lot of descriptive parts could be minimized, and the focus be given to the functional possibilities/relationships of the specific mutant lines to the metabolic pathway(s) impacted. This can then be translated into the significance of these studies for nutrition and other aspects of ripening processes. For instance, what evidence is there for the authors to suggest that the enhanced ripening causes membrane breakdown and thereby release of some of the molecules the authors quantified, A deeper insight of the data presented is missing.

In its present narrative, this manuscript is purely descriptive and lacks scholarship in terms of condensed thinking about metabolic regulation vis a vis each mutation.”

As requested by the Reviewer, the Discussion section (now L408-540) has been completely re-written and re-elaborated. We now present the effect of each mutation in separate sub-sections trying to make clear the general impact on fruit composition, with emphasis on molecules involved in organoleptic and nutritional quality. This was aimed to find the “scholarship of synthesis of information” that was missing. Seven new references have been cited to improve this aspect.

Reviewer 2 Report

The submitted manuscript titled "Color Mutations Alter the Biochemical Composition in the San Marzano Tomato Fruit" by Gabriella Dono et al. is a scientifically sound research article that aims to characterize the volatile and non-volatile compounds of six tomato lines introgressing three tomato mutations, either in single or double combination. The characterization of the biochemical effects of these mutations was based on the comparison with the SM original parent, with the final aim of obtaining new mutants with diversified nutritional, aesthetic and flavor characteristics.

The article is very well organized and divided into subsections, which makes reading easier. The work is written very carefully, and contains an interesting discussion. Although, I would be happy to recommend this manuscript for publishing, there are few minor issues, that should be improved:

  • Please include in the introduction the information how different biochemical composition of tomato fruits affects the nutritional value
  • Page 4, line 110: “One mL” should be replaced by “1 mL”
  • Page 7, Fig2. The legend “a” and “b” are missing. I recommend to add the additional legend with the colors of dots and corresponding explanations directly in the figure 2b, instead of description “Color symbols in the loading plot (b) correspond to benzenoids (B, green), branched-chain amino acid related (BCAA, blue), apocarotenoids (C, red), esters (E, yellow), fatty acid derivatives (L, light blue), phenylalanine derivatives (Phe, orange), sulphur compounds (S, brown), monoterpenoids (T, water green).
  • Page 10, Fig. 4. Please see the comment above
  • Page 13, Fig 7. The legend “a”, “b” and “c” should be above (not below) the venn diagrams
  • The style of references should be corrected and unified

Author Response

Reviewer n. 2

“Please include in the introduction the information how different biochemical composition of tomato fruits affects the nutritional value”

Few sentences (L44-55) have been integrated to address this point, including four new references.

“Page 4, line 110: “One mL” should be replaced by “1 mL””

We corrected accordingly (L122).

“Page 7, Fig2. The legend “a” and “b” are missing. I recommend to add the additional legend with the colors of dots and corresponding explanations directly in the figure 2b, instead of description “Color symbols in the loading plot (b) correspond to benzenoids (B, green), branched-chain amino acid related (BCAA, blue), apocarotenoids (C, red), esters (E, yellow), fatty acid derivatives (L, light blue), phenylalanine derivatives (Phe, orange), sulphur compounds (S, brown), monoterpenoids (T, water green).”

Symbols “a” and “b” for single graphs have been introduced as well as the Legend in Figure 2b. Figure legend has been accordingly modified (L254-257).

“Page 10, Fig. 4. Please see the comment above”

Legends in Figure 4b,d have been introduced and figure legend accordingly modified (L334-336).

“Page 13, Fig 7. The legend “a”, “b” and “c” should be above (not below) the venn diagrams”

The figure has been corrected accordingly.

“The style of references should be corrected and unified”

References have been checked and corrected for style.

Reviewer 3 Report

The manuscript of Dono et al presents the differences in the metabolic profiles of transgenic lines of the San Marzano tomato variety. The topic is interesting and is well-presented. However, I feel that the scope of this research is not clearly presented in the manuscript. Moreover, the authors should describe with more detail the quality and organoleptic attributes of the SM variety in order for the reader to obtain a more clear view of the effects of the treatments on these characteristics. My final comment is that I have found the supplementary tables more useful than some of the figures, however, given the size of the required revision I will not advice for any changes. 

My recommendation to the editor is to accept the manuscript after a minor revision from the authors.  

Author Response

Reviewer n. 3

“I feel that the scope of this research is not clearly presented in the manuscript. Moreover, the authors should describe with more detail the quality and organoleptic attributes of the SM variety in order for the reader to obtain a more clear view of the effects of the treatments on these characteristics.”

To better highlight the scope of the research, in Introduction a sentence has been added (L52-55). The whole Discussion section (now L408-540) has been re-written and re-elaborated with the aim of better summarize the extent of biochemical variation given by each mutation, that was the scope of the research.

In addition, the quality attributes of SM have been better detailed in Introduction (L77-83) and two new references have been added.

Round 2

Reviewer 1 Report

The Discussion section has been improved in the revised version.